# Sports Utility Vehicles: A Public Health Model of Their Climate and Air Pollution Impacts in the United Kingdom

**DOI:** 10.3390/ijerph20116043

**Published:** 2023-06-02

**Authors:** Charles Dearman, James Milner, Glenn Stewart, Giovanni S. Leonardi, John Thornes, Paul Wilkinson

**Affiliations:** 1Centre for Climate and Health Security, UK Health Security Agency, London SW1P 3HX, UK; 2Department of Public Health, Environments and Society, London School of Hygiene and Tropical Medicine, London WC1E 7HT, UK; 3Centre on Climate Change and Planetary Health, London School of Hygiene and Tropical Medicine, London WC1E 7HT, UK; 4Department of Public Health, London Borough of Enfield Council, London EN1 3XA, UK; 5Environmental Epidemiology Team, UK Health Security Agency, Chilton OX11 0RQ, UK

**Keywords:** sports utility vehicle, climate change, public health, air pollution, nitrogen dioxide, carbon dioxide, transport, electric cars, electrification

## Abstract

The emission benefits of shifting towards battery electric vehicles have so far been hampered by a trend towards sports utility vehicles (SUVs). This study assesses the current and future emissions from SUVs and their potential impact on public health and climate targets. We modelled five scenarios of varying SUV sales and electrification rates, and projected associated carbon dioxide (CO_2_) and nitrogen oxide (NO_x_) emissions. Multiple linear regression was used to determine the relationship between vehicle characteristics and emissions. Cumulative CO_2_ emissions were valued using the social cost of carbon approach. Life table analyses were used to project and value life years saved from NO_x_ emission reductions. Larger SUVs were disproportionately high emitters of CO_2_ and NO_x_. Replacing these with small SUVs achieved significant benefits, saving 702 MtCO_2_e by 2050 and 1.8 million life years from NO_2_ reductions. The largest benefits were achieved when combined with electrification, saving 1181 MtCO_2_e and gaining 3.7 million life years, with a societal value in the range of GBP 10–100s billion(s). Downsizing SUVs could be associated with major public health benefits from reduced CO_2_ and NO_x_ emissions, in addition to the benefits of electrification. This could be achieved by demand-side mass-based vehicle taxation and supply-side changes to regulations, by tying emission limits to a vehicle’s footprint rather than its mass.

## 1. Introduction

Internal combustion engine (ICE) cars are major drivers of climate change and air pollution, both of which harm public health [1]. Climate change is considered one of the greatest threats to global public health and requires urgent adaptation and mitigation responses [2]. The United Kingdom (UK) has committed to achieving net zero, i.e., zero greenhouse gas (GHG) emissions or balanced by removal from the atmosphere, by 2050 under the Climate Change Act 2008 [3]. Decarbonising road transport is vital to achieving this, as, in 2021, it was responsible for 23.5% of net GHG emissions, the highest of any sector [4].

Similarly, transport-related air pollutants are known to be a major determinant of ill-health worldwide [5], with increased health risks, including respiratory, cardiovascular and neoplastic diseases [6]. Exposure to nitrogen dioxide (NO_2_), a product of fossil fuel combustion, is thought to have a causal detrimental impact on health [7,8]. In 2015, the UK Government estimated that long-term NO_2_ exposure increased mortality by up to 23,500 deaths per year nationally [9]. Following repeated NO_2_ concentration limit breaches, subsequent legal action [10,11] and public anger following the diesel emission scandals [12], controlling NO_2_ remains a policy priority [13]. In 2021, road transport was responsible for 27.3% of UK annual nitrogen oxides (NO_x_) (a collective term for nitric oxide and NO_2_ emissions), again the largest contributing sector [14].

Technological advances have enabled the implementation of policies aiming to decarbonise transport, such as the UK Government plan to ban ICE passenger vehicles by 2035 [15]. This policy could realise major co-benefits for public health by also reducing air pollution. Battery electric vehicles (BEVs) are expected to be the dominant replacement for ICE vehicles [15] and have zero tailpipe emissions. However, whilst a transition to BEVs would generally be expected to reduce CO_2_ and air pollution emissions, there are several limitations and caveats [16]. For example, the overall environmental health impacts of BEVs depend on both the source of electricity [17,18,19] and pollution relating to the production of components [17,18,19,20]. There is also uncertainty regarding non-tailpipe emissions (brake, tyre, and road surface wear particles), which could worsen with the increased weight of BEVs [21,22] or conversely improve due to regenerative braking and improved tyre design [23]. Despite these issues, a net mortality benefit of electrification has been projected in France [24], the United States [25,26], and Australia [27]. In California, BEVs have already been associated with improved air quality [28], with related possible morbidity improvements [29].

However, these potential benefits of electrification have been negated by the increasing market share of heavier vehicles, especially sports utility vehicles (SUVs) [30]. In 2019, SUVs accounted for over 44% of car sales globally, up from 20% in 2010, with a similar trend seen in the UK [31]. The International Energy Agency (IEA) estimates that SUVs were the second largest cause of energy-related GHG emission growth over the last decade (after electricity generation) [31]. This GHG emission growth more than offset savings in improvements in energy efficiency and electrification of vehicles. The popularity of SUVs has risen due to consumers’ preference for a high-riding position, style, perceived and real utility gains, and strong advertising by auto-manufactures, reflecting the higher profit margins for SUVs over other segments [32,33,34]. As SUVs are typically heavy, large, and powerful vehicles, they tend to have increased inertia and rolling resistance [35,36,37] and worse aerodynamics [35,36,37]. As a result, they tend to have poorer fuel efficiency than smaller and lighter vehicles of the same fuel type [38]. This would be expected to increase tailpipe emissions and be at odds with the government’s ambitions to address air pollution and climate change.

Despite the potential for the growth of SUVs to undermine decarbonisation and air quality policies, this issue has not been extensively investigated from a public health perspective. Investigations into the health impact of SUVs have instead concentrated on safety in collisions. They suggest a marked increased risk of mortality for pedestrians, cyclists [39,40,41], and occupants of conventional cars when involved in collisions with an SUV [42,43]. To our knowledge, there are no peer-reviewed studies in the literature assessing the public health impact of SUVs from their tailpipe emissions. Whilst health impact assessments from abated particulate matter have been conducted on multisectoral UK climate mitigation pathways [44], the health impact of the UK’s planned phase out of new ICE vehicles by 2035 has not been studied in isolation or with regard to NO_x_. Understanding the impact of SUVs is of urgent importance due to the rapidly increasing popularity of SUVs at the same time as the need for bold decarbonisation of the sector.

This study aims, for the first time, to model the current and future public health and climate impact of SUVs, in relation to NO_x_ and CO_2_ emissions, to inform UK decarbonisation and air pollution policies. To achieve this aim, the novel objectives of the study are as follows. First, we aim to determine the absolute contribution of SUV tailpipe NO_x_ and CO_2_ emissions, and their relative contribution to the total UK passenger vehicle fleet emissions. Second, we aim to investigate the relationship between SUV characteristics (such as vehicle mass) and CO_2_ and NO_x_ emissions. Third, we aim to explore how passenger vehicle CO_2_ and NO_x_ emissions could differ by 2050 by projecting various policy scenarios that change SUV sales during the phase out of ICE vehicles. Fourth, we aim to quantify the relative potential public health impacts between projections. Finally, this study will also estimate, for the first time, the public health impact of changes in NO_2_ from the UK policy to phase out ICE passenger vehicles by 2035.

## 2. Materials and Methods

### 2.1. Conceptual Framework

This study uses a bottom-up approach to model the current and future tailpipe emissions of the UK passenger vehicle fleet under various scenarios. The model is built from data on the emission profiles of individual cars registered for road use in the UK. The CO_2_ and NO_x_ model have differing aims (to compare cumulative CO_2_ emissions and to inform life table health calculations) and, therefore, differ in their conceptual construction. The scenarios are designed to mimic a diversity of possible policy approaches. Multiple linear regression is used to determine which vehicle characteristics may explain the differences in emissions between policies.

The CO_2_ model assesses the climate impact of new cars added to the UK fleet overtime from the end of 2020. The model does not measure emissions from the existing UK fleet. This is because it is the cumulative CO_2_ that is generated following the introduction of a policy (in this model introduced at the end of 2020) that affects new car sales in which we are interested. For example, this could be a policy switching from larger to smaller SUVs, or a proportional replacement of ICE vehicles with BEVs. This model allows the interrogation of the relative performance of a policy as a climate mitigation measure, at any point in the future. The CO_2_ model controls for vehicle stock turnover from 2020 onwards, using historical data patterns. It also includes the carbon emitted from the generation of electricity used to power BEVs in the fleet.

The NO_x_ model is more complicated as emissions from the entire UK fleet need to be calculated to project relative atmospheric changes in NO_2_. First the NOx emissions for all passenger vehicles registered in previous years are calculated. Older vehicles produce significantly more NO_x_ as they would have complied with earlier and less stringent European (EURO) emission standards. Then, the NO_x_ emissions of new vehicles each subsequent year are added whilst older vehicles are phased out using the same stock turnover model as for the CO_2_ model. The changes in total fleet NO_x_ emissions can then be compared to a baseline of business as usual and translated to proportional changes in atmospheric NO_2_ concentrations. The change in atmospheric NO_2_ concentrations can then inform life table calculations to calculate the public health impact.

### 2.2. Data Handling and Multiple Linear Regression

All passenger vehicle models with 250 or more new registrations recorded by the Driver and Vehicle Licensing Agency (DVLA) in the fourth quarter (Q4) of 2020 were selected for analysis (dataset VEH0161) [45]. Data on the real driving emissions (RDE) test for NO_x_ emissions by make and model of vehicle was sourced from the UK’s Vehicle Certification Agency [46]. Vehicle characteristics and worldwide harmonised light vehicles test procedure (WLTP) CO_2_ emissions were sourced from the European Environment Agency (EEA) database on all new passenger vehicles registered in the EU and Great Britain [47]. Data extracted included kerb weight, WLTP test weight, fuel type, engine capacity, wheelbase length, and WLTP CO_2_ emissions for make, as well as specific model variants. The bestselling variant of any generic car model was assumed to be indicative of all variants sold. These three data sources were manually combined so that for each vehicle model, its characteristics, pollutant profile, and number of registrations were known.

Each vehicle model in the database was assigned a EURO segment category according to the manufacturer’s website (examples in Table 1). The SUV segment was sub-categorised by mass, with small defined as <1500 kg (similar to A, B, and C segments), medium as 1500–2000 kg (similar to D, E, and F segments) and large SUVs as >2000 kg, to reflect the diverse vehicle models within this segment.

The relationship between vehicle CO_2_ or NO_x_ emissions and vehicle characteristics (mass, fuel type, engine capacity, and wheelbase) was characterised by multiple linear regression using the regress command in Stata17^®^ (StataCorp, College Station, TX, USA).

### 2.3. Scenarios

We modelled five scenarios designed to represent contrasting levels of SUV sales and fleet electrification. All scenarios assume that 2021 sales return to 2.5 million units (pre-pandemic average for 2016–2019) following the COVID-19 pandemic impact on new vehicle sales in 2020, then remain constant into future years, in line with approximate future industry expectations [48]. Scenarios were extended into the future until a steady state of emissions was achieved for all scenarios.

Scenario 1 represented business as usual (BAU): EURO segments proportions, fuel types, and fuel efficiency for new passenger vehicles registrations remain constant from 2020. In Scenario 2, the ‘Small SUV’ scenario, medium and large SUV sales are replaced exclusively by small SUV sales from 2020 on a 1:1 basis. It incorporated the same assumptions as Scenario 1.

In contrast for Scenario 3, the ‘No SUV’ scenario, small, medium, and large SUVs are replaced by segment B, C, and D average cars, respectively, on a 1:1 basis. This is to reflect the fact that SUVs are often bought for increased space and that these segments maintain similar internal space but without the high-riding positions characteristic of SUVs.

The next two scenarios incorporated electrification plans. Scenario 4, ‘Electrified’, is consistent with UK Government policy to ban exclusively diesel or petrol passenger vehicles (but not plug-in hybrids, or PHEVs) by 2030 and to ban PHEVs by 2035. This model is based on the Society of Motor Manufacturers and Traders’ projections that BEV sales will represent 25% of the market in 2025, 70% by 2030, and 100% by 2035 [48]. It is assumed that the diesel and petrol proportions of the non-electrified sales remain constant from 2020 until 2030. From 2030–2035, non-fully electrified vehicles are assumed to all be petrol PHEVs, reflecting the negligible number of diesel PHEVs currently in the UK fleet (<2%) and the continued fall in diesel popularity in recent years. Hydrogen cars are not considered as only two models are available in the UK, with a total of 200 vehicles licensed on UK roads in 2020 [45].

Scenario 5, ‘Electrified and small SUV’, envisages the gradual electrification pathway of scenario 4 combined with the immediate and total replacement of new large and medium SUVs with small SUVs seen in scenario 2. This scenario aims to estimate the public health benefit of reducing the mass of SUVs without fundamentally changing their aesthetic as an extension to the UK Government’s electrification policy.

### 2.4. CO_2_ Emission Calculations

All petrol cars were assumed to drive 10,137 km and diesel cars 15,125 km per annum, using the mean estimates for England, according to the National Travel Survey 2019 [49]. The annual distance driven for electric cars was assumed to be 15,180 km based on an estimate of the Royal Automobile Club Foundation [50]. Similar data were not publicly available for other UK nations, so it was assumed to be the same across the UK in this analysis. CO_2_ emissions were calculated for each vehicle model in 2020 using the following equation.

Equation (1) is as follows:C × R × D/1,000,000 = T(1)
where:
C = WLTP CO_2_ emissions (g/km);R = number of model vehicle registrations in 2020;D = average distanced travelled for fuel type (km);T = Total annual CO_2_ emissions for model sold in 2020 (tonnes).


T was then summed over all models to calculate the total emissions for all models of a segment to give segment emissions. Average CO_2_ emissions per vehicle by segment was calculated by dividing total segment CO_2_ emissions by segment vehicle count. This then gave an emission value that could be used in modelling changing vehicle segment sizes over time. Future emissions from vehicles registered in previous years were included, and gradually reduced as the vehicles were phased out, according to patterns of stock replacement in 2020 (which indicate a median vehicle lifespan of 14 years).

BEVs have zero tailpipe GHG emissions but have indirect emissions from the electricity used to power the vehicles. These can be calculated by the carbon intensity of the grid from which BEVs are charged. The future carbon intensity of the UK grid in g/kWh was calculated from the Climate Change Committee’s ‘*Balanced Net Zero Pathway*’ projections for electricity generation and electricity sector CO_2_e emissions to 2050 [51]. It is assumed that all BEVs had a fuel efficiency of the best-selling BEV in the UK in 2020, the Tesla Model 3, of 6.5 km/kWh [52]. This value was then applied to the average annual carbon intensity of the UK grid to give a predicted gCO_2_e/km for BEV adjusted for the expected continued decarbonisation of the electricity sector until the year 2050, using the following formula.

Equation (2) is as follows:

Annual CO_2_e emissions for BEV:E × I_year_ × D/1,000,000 = T(2)
where:
E = efficiency (set at 6.5 km/kWh for BEVs);I_year_ = mean annual electric grid carbon intensity of that year (gCO_2_e/kWh);D = distance travelled (set at 15,180 km for BEVs);T = annual CO_2_e (tonnes).


Total CO_2_e emissions for a BEV could then be calculated by summing the annual CO_2_e for a vehicle’s lifetime (median 14 years) or a specific time point (2035 and 2050 in this analysis).

For example:T_2020_ + T_2021_… + T_2033_ = L_2020_(3)
where L_2020_ is the lifetime emissions for a BEV registered in 2020 reported at the end of 2035 or 2050 (last year of service would be 2033), whilst for a BEV registered in 2049:T_2049_ + T_2050_ = L_2049_(4)
where L_2049_ is the lifetime emissions for a BEV registered in 2049 reported at the end of 2050.

The lifetime emissions of a BEV could then be multiplied by the number of registrations of BEVs in that year to give lifetime emissions of BEV fleet of any given year.

For example:L_2020_ × R_2020_ = F_2020_(5)
where:
R = number of vehicle registrations;F_year_ = Total BEV fleet emissions for that year.


The total CO_2_e from BEVs at a given time point (e.g., 2050) in a given scenario could then be given by:F_2020_ + F_2021_…+ F_2050_ = Total CO_2_e by 2050(6)

CO_2_ changes were valued using the UK Government’s marginal CO_2_e abatement projected costs, which estimate the value that society places on 1 tonne of CO_2_e emitted for a given year. Low, central, and high estimates are published to reflect underlying uncertainties and are designed to be consistent with the marginal cost of meeting net zero by 2050 target [53].

### 2.5. NO_x_ Emission Calculations

Modelling annual NO_x_ emissions of the UK fleet requires calculating the emissions from the existing stock plus new vehicles added that year, minus vehicles decommissioned. To accomplish this, a stock turnover model was created based on historic patterns of UK vehicle lifetimes (by combining DVLA datasets VEH0124 [45] and VEH1153a [45]). to calculate the yearly NO_x_ emissions of existing vehicles on the UK roads that had been registered in previous years. This allowed the NO_x_ emission contribution of the pre-2020 UK fleet to be calculated for future years as the vehicles are slowly phased out, assuming they emit the corresponding maximum EURO emission standard legal limit of their first year of registration. This also provided a means to estimate the lifespan of future vehicles in the model. It suggested that, for a policy applied to new vehicles from 2020 onwards, it would cover 41% of the total UK fleet by 2025, 77% by 2030, and 96% by 2035. The implication of this is that any policy introduced for new cars would take many years to have a its full impact, because of the approximately 14-year median lifespan of a UK passenger vehicle. It is important to note that this is even slower for electrification, which is a transition rather than an abrupt policy change.

The contribution from new vehicles can be added each year to this model. Annual predicted NO_x_ emissions for a vehicle model registered in 2020 were calculated using Equation (7), as follows:N × R × D/1,000,000,000 = T(7)
where:
N = RDE NO_x_ emissions (mg/km);R = number of registrations of vehicle model in 2020;D = average distanced travelled for fuel type (km);T = Total annual NO_x_ emissions for vehicle model sold in 2020 in first year of use (tonnes).


The mean NO_x_ emissions per vehicle by segment were calculated by summing T for all models of a segment and then dividing by the total number of registrations in that segment. This was used to model future scenarios in the same fashion as in the CO_2_ model above. Future ICE cars were assumed to continue to produce the same mass of NO_x_ as in 2020.

The relative change in NO_2_ for each scenario relative to BAU was calculated by assuming that under the BAU scenario the urban background concentration of NO_2_ from UK passenger cars would remain constant at 2.59 mg/cm^3^ [54]. As a result, the proportional change relative to the BAU scenario could be applied for each year to calculate the projected change in NO_2_ in mg/cm^3^. The absolute changes in NO_2_ concentrations then informed the life table calculations (below).

### 2.6. Life Table Analysis for NO_x_ Scenarios

The Institute of Occupational Medicine’s IOMLIFET (version 2013) package of Microsoft Excel™ spreadsheets was used to perform life table calculations to quantify the predicted impact on mortality of changes in NO_2_ exposure under each scenario [55]. The life table models change in population survival over time, resulting from changes in underlying mortality rates. The strength of IOMLIFET is that it allows for changes in air pollutant coefficients each year to be combined with lagged effects of air pollutants from previous years. It calculates separate hazards for each future year and age group.

The life table model was populated with UK population data for 2019 by sex and year of age [56], as well as counts of deaths by each year of age and sex [57] (the model assumed the hazards are the same for ages 90–105 inclusive). Data from 2019 were used instead of 2020 because of the increased number of deaths during the COVID-19 pandemic in 2020. This provided an all-cause annual baseline hazard of death for each year of life and by sex. The annual population, birth rate, and baseline hazard of 2019 were kept constant into future years. The main outputs were life years gained by 2126 and the total value of the life years gained. This study used the current UK Government guidance recommended value of a life year (VOLY) of GBP 60,000 per year and a discount rate of 1.5% for health-related interventions [58].

This study used the unadjusted NO_2_ coefficient of 1.023 (*95% CI: 1.008*, *1.037*) per 10 μg/m^3^ as per the UK’s Committee on the Medical Effects of Air Pollutants (COMEAP) guidance ‘*to assess the health benefits of interventions that reduce a mixture of traffic-related pollutants*’ [8].

This means that, for a given NO_2_ reduction (∆), a new relative risk of mortality was calculated for each age group and for each year, into the future, using the following Equation (8):Relative risk = 1.023^(−∆/10)^(8)

The relative risk was multiplied by the baseline mortality rate to calculate the mortality rate. From this, survival probability, cumulative survival, and expected life years can be calculated [55]. Relative risks were applied at all ages of 30 years and above, reflecting a lack of evidence of air pollution effects in those younger than 30 and common practice for air pollution mortality calculations [7]. A sensitivity analysis was conducted using alternative values of VOLY and discount rates commonly used in the UK [58,59] and the upper and lower 95% confidence interval unadjusted NO_2_ coefficient limits.

It is expected that a change in concentration of an air pollutant (in this analysis, background urban annual mean NO_2_ concentrations) will be followed by a lag before mortality impacts are observed. Given a lack of empirical evidence, lag functions for air pollution are generally derived from expert consensus rather than experimentally. We applied the US Environment Protection Agency (EPA) proposed lag structure [60], which sees 30% of the effect seen within the first year and the full effect seen after 20 years.

All data used are from secondary sources fully in the public domain, and the study was assessed by the London School of Hygiene and Tropical Medicine Research Governance and Integrity Office as not requiring ethical approval.

## 3. Results

### 3.1. Vehicle Mass Impact on CO_2_ and NO_x_ Emissions

ICE SUVs were disproportionately represented in heavier segments of the UK fleet, making up 63% of vehicles of 2000 kg and above, 38% from 1500–1999 kg, and only 35% of those below 1500 kg (Figure 1a).

There was a positive correlation, with a broadly linear relationship, between vehicle mass and CO_2_ emissions for diesel, petrol, and petrol hybrid vehicles (Figure 1b), in keeping with other analyses [31]. The CO_2_ model consisted of 181 vehicle models, equating to 1,560,452 out of 1,656,403 (94.2%) 2020-registered passenger vehicles (Table 2).

Overall, 61.5% of the vehicles included were petrol, 20.3% were diesel (reflecting the rapid decline in diesel sales since 2016), and 4.7% were BEVs. The most popular segment was SUVs (J segment) at 40.2%, followed by small cars (B segment) at 22.9%. Multiple linear regression analysis suggested that vehicle mass was a strong predictor of CO_2_ emissions (Table 3). All else being equal, a 100 kg increase in vehicle weight was associated with a CO_2_ emission increase of 10.3 g/km for ICE passenger vehicles.

There was a similar but weaker correlation between ICE vehicle mass and NO_x_ emissions. The NO_x_ emission model was created from RDE values available for 985,560 vehicles registered in 2020 (59.5% of the total). This lower coverage than the CO_2_ model was due to the absence of submitted data by the Volkswagen Group and Bayerische Motoren Werke (BMW) auto-manufacturers at the time of analysis. Multiple linear regression showed mass and engine capacity were predictors of NO_x_ emissions (Table 4), but not as strongly as for the CO_2_ analysis. All else being equal, a 100 kg increase in vehicle mass is associated with an RDE NO_x_ emission increase of 6.7 mg/km for diesel vehicles and a 7.4 mg/km increase for petrol.

The results of the regression analysis suggests that policies that transition away from heavy ICE vehicles (e.g., larger ICE SUVs) to lighter vehicles could be associated with significant CO_2_ savings. The model showed that SUV vehicles contributed most to the overall CO_2_ emissions of the 2020 registered fleet in the UK and that diesel vehicles are more common in the larger and heavier vehicle segments (J and M). The large proportion of CO_2_ emissions from SUVs is due, firstly, to the high popularity of SUVs (Figure 2b) and secondly, to the high per unit CO_2_ emissions for large and medium SUVs (Table 5). This data suggests the mean large SUV emits 127% (1.6 tonnes) more CO_2_ annually than the mean A segment vehicle per year.

### 3.2. Scenario Projections for CO_2_ and NO_x_ Emissions

Scenarios incorporating the transition from larger to small SUVs have an immediate and major annual CO_2_ saving, and those that include electrification have a more gradual saving (Figure 3a). The ‘Small SUV’ scenario achieves CO_2_ emission reductions greater than the UK Government’s ‘Electrified’ scenario by 2035, but the benefit of the ‘Electrified’ scenario is larger by 2050 (Figure 3b). However, there is a 16% added benefit in incorporating an immediate shift to smaller SUVs versus electrification alone by 2050 (Table 6). To put this in context, the saving for the ‘Electrified and Small SUV’ scenario versus the BAU of 1181 MtCO_2_ from 2020 to 2050, is equivalent to 2.2 times the GHG emissions of the entire UK economy for the year 2019 [61].

The CO_2_ emissions associated with charging BEVs were also included in these calculations but were found to be insignificant, reducing the emissions savings by less than 1% by 2050 in electrification pathways.

The modelling for NO_x_ changes was more complex given the need to project atmospheric changes (Figure 4). Figure 4a shows the expected NO_x_ emissions of passenger vehicles in their first year of use from 2020–2035 under the five scenarios and shows a similar pattern to the CO_2_ model. Figure 4b shows the annual NO_x_ emissions for all UK passenger vehicles, which shows how emissions fall in all scenarios (due to the phasing out of more polluting older cars) but with the greatest reductions in electrified pathways (reaching negligible tailpipe NO_x_ emissions by 2051). Figure 4c shows how these changes translate to reductions in urban atmospheric concentrations of vehicle-related NO_2_ pollution relative to BAU, assuming direct proportionality between the reduction in NO_x_ and NO_2_.

### 3.3. Health and Monetary Impacts

The marginal abatement costs of making equivalent CO_2_ emission reductions to those in the scenarios in 2035 or 2050 are shown in Table 7. The central estimate suggests that UK society would be willing to pay up to GBP 447 bn in 2050 to achieve the same carbon savings as projected by the electrified and small SUV scenario implemented from 2020.

All scenarios were associated with gains in life years (i.e., positive health benefits) compared to business as usual, with the largest gains for the electrified and small SUV scenario of approximately 3.7 million life years. The impact of inception lags was small and associated with a decrease in the life years gained of 4–5% (as such lagged estimates are used in all data shown).

By applying the UK Government (HM Treasury) recommended discount rate for health-related interventions of 1.5% and value of VOLY of GBP 60,000 to the lagged estimates of life year gained, reasonable summary estimates of the scenarios can be made (Table 8). The results suggest the that ‘Electrified and Small SUV’ scenario has the greatest number of life years gained and associated value. Interestingly, because the benefit of the change to small SUVs occurs sooner than the benefit of electrification, the monetary value is proportionally greater than the life years gained (9.1% versus 5.4%).

The model was sensitive to the choice of NO_2_ coefficient (Figure 5), VOLY value, and discount rate (demonstrated in Table 9 for the most ambitious scenario, ‘Electrified and Small SUV’). Overall, this showed that the value of life years gained was highly dependent on the NO_2_ coefficient used, as well as the discount rate. The value assigned to a single life year had a proportional multiplicative effect on total value.

## 4. Discussion

Modelling studies, such as this one, can only provide broad estimates of impacts, but our study suggests that large and heavy SUVs are detrimental to the climate (through their greater CO_2_ emissions) as well as to health (through excessive NO_x_ emissions). When combined with the popularity of SUVs, the model suggests the SUV segment emits more CO_2_ and NO_x_ pollution than any other, which is in keeping with other analyses [31]. Importantly, this study shows that medium and large SUVs, which are heavier and more likely to be diesel, are responsible for the majority of the SUV segment’s emissions.

Vehicle mass was shown to be the key predictor of CO_2_ emissions and a significant but weaker predictor of NO_x_ emissions, adding to previous work showing the importance of mass to fuel efficiency [35,36,37,62]. Increasing mass increases the inertia and rolling resistance that must be overcome to accelerate a vehicle and, therefore, increases the energy expended. This study did not investigate the role of potential increased aerodynamic drag for SUVs. However, this is only a significant form of energy loss at higher speeds, such as motorway driving, and previous studies have shown vehicle mass to be a more significant cause of energy loss during typical car journeys [36] and under controlled driving cycles [37].

For CO_2_ emissions, the model suggests that promoting lighter vehicles, even if they are still SUVs, would offer major emission benefits (‘Small SUV’ scenario). The emissions averted are largest when lighter vehicles are combined with electrification (‘Electrification and Small SUV’ scenario) and provide a substantial additional benefit. This is an important finding as it emphasises that the overall impact of the electrification transition can be significantly improved by incentivising switching to smaller vehicles. The cumulative CO_2_ difference between the ‘Electrified’ and ‘Electrified and Small SUV’ scenarios from 2020–2050 is 225 Mt. To give that context, 225 MtCO_2_e is equivalent to 55% of the UK’s entire CO_2_e emissions for the year 2020 [63].

For NO_x_ emissions, large public health benefits were projected by switching to smaller vehicles. The ‘Small SUV’ scenario resulted in immediate and major improvements in NO_x_ emissions, with a central estimate of around 1.8 million life years gained. The greatest benefits in NO_2_ reduction and life years were achieved, when this was combined with electrification, with a central estimate of 3.7 million life years gained.

This study has suggested that, in the UK, a shift from heavier SUVs to lighter and smaller vehicles, even if still classed as an SUV, would lead to benefits for the climate and health. Importantly these benefits could be realised more quickly than and in addition to those expected by electrification alone. Such vehicle downsizing achieves greater fuel efficiency by reducing energy loss to inertia (by reducing mass), rolling resistance (using thinner tyres and reduced mass), and aerodynamic drag (with a smaller frontal area). An alternative is light-weighting, which describes the use of lighter materials to produce a vehicle of the same dimensions, which would reduce energy loss by overcoming inertia and rolling resistance.

### 4.1. Policy Implications

The findings suggest significant societal benefits in moving away from heavy SUVs, in addition to electrification, with immediate policy implications. This study adds nuance to the current policy priority of electrification, highlighting that the vehicle segment, as well as fuel type, is important in meeting our climate and air pollution goals.

The ‘Small SUV’ scenario mimics the introduction of a policy with an immediate switch to small SUVs compared with the graduated transition to BEVs in electrified scenarios. Although this is a simplification, it is not entirely unrealistic, as consumer preference for cars can shift suddenly. For example, from 2016–2021, new diesel registrations collapsed by 85% in the UK [45]. A similar shift to lighter SUVs could occur in the UK, if incentivised, as most consumers seek the SUV aesthetic and high-riding position but most do not require the added utilities of the larger, more powerful SUVs, such as off-roading capability [31]. The ‘Small SUV’ scenario achieves greater reductions than electrification policies by 2035. This improved rate of change is important considering current UK air quality plans have been found to be illegal under UK law [11] and earlier climate mitigation is more effective than later [2].

A further benefit of policies promoting lighter SUVs is the reduced relative disruption compared to electrification. To achieve the UK Government’s electrification transition there must be major changes to society (e.g., consumer education, charging stations, electrical grid improvements) and the auto-manufacturer industry (research and development, supply lines, and retooling factories) [64]. These changes are likely to take longer than incentivising a shift from heavy to lighter SUVs. This could occur from supply-side regulatory changes and demand-side interventions, such as tax incentives.

Regarding supply-side measures, a principle aim of auto-manufacture regulators is to promote vehicle efficiency whilst maintaining a diversity of vehicles that suit different needs, such as varying passenger or cargo capacity [65]. To achieve this in the European market, CO_2_ emission limits for individual manufacturers are adjusted by the average mass of the manufacturer’s fleet [38]. This allows heavier fleets to have disproportionately higher CO_2_ emission targets. In practice, targets between manufacturers differed by up to 11 g/km in 2021 [38]. As lighter vehicles are subject to stricter CO_2_ limits, it can lead to manufacturers designing heavier cars to avoid these limits [38], introducing a perverse incentive towards heavier vehicles. This is then compounded by the less stringent emission targets for heavier vehicles, allowing higher power (and often diesel) engines to be fitted, worsening emissions [65]. This policy further accentuates the correlation of vehicle mass and CO_2_ emissions due to the role of mass in inertia and rolling resistance forces.

The flaw in this regulation is choosing a proxy measure of utility (vehicle mass) that is also an independent predictor of CO_2_ emissions [66]. A better utility parameter might be vehicle footprint, calculated by multiplying wheelbase by track width (i.e., the area between the four points where the wheels touch the ground [67]), which is not directly related to CO_2_ emissions. This should incentivise the use of light-weighting to reduce CO_2_ emissions. A further benefit of light-weighting could be the use of less material and, therefore, reduced supply chain-related emissions, although this would be highly dependent on the materials and manufacturing process.

For demand-side measures, several European countries use vehicle tax to adjust consumer behaviour towards actions to promote decarbonisation and reduce air pollution, but only a few countries link the tax to vehicle mass. In the UK, new passenger vehicles must pay a graduated ‘vehicle emissions duty; based on CO_2_ emissions but not mass. In contrast, French vehicle tax is designed to penalise high CO_2_ emitting vehicles and has additional tax for cars over 1800 kg, charged at a rate of €10/kg [68]. In Norway, which boasts the highest BEV market share of any country, ICE vehicles must pay additional taxes for CO_2_, NO_x_, and mass, whilst BEVs benefit from a range of incentives [69,70]. The policies of France and Norway show that it can be politically and practically feasible to tax vehicle mass to achieve both CO_2_ and air quality aims.

Such policy changes could be supported by further research, such as extending the modelling to other pollutants, especially particulate matter, including from non-tailpipe sources, and considering morbidity as well as mortality [71].

### 4.2. Strengths and Limitations

This study demonstrates a novel way to combine publicly available databases to interrogate the climate and health impacts of a specific vehicle segment. This is the first study to model the NO_x_ pollution and resultant population health effects of SUVs. For CO_2_, whilst it was known that SUVs were high emitters, this is the first study to attempt to quantify annual emissions for a national fleet and to model this in relation to electrification and decarbonisation policies and then assess the resulting health impact.

The study used the best available but imperfect data and the model required assumptions and simplifications that could both over- and underestimate the findings. For example, there are well recognised limitations in the measurement of tailpipe emissions. The use of WLTP-derived CO_2_ emission data is known to be overly optimistic of real-world driving [72]. For NO_x_ emissions, independent roadside testing suggests that using WLTP reported data is also likely to underestimate emissions [73]. However, no single test can describe real-world driving emissions which are dependent on multiple highly variable factors (e.g., driving style, road conditions, temperature, vehicle condition, and loads).

The sensitivity analysis showed that the estimation of the public health impact of NO_2_ was sensitive to the chosen exposure–response coefficient (Figure 5), the discount rate and monetary value assigned to a life year. The estimates were not particularly sensitive to the use of an inception lag. This study did not adjust for rebound effects that often result from energy efficiency interventions. A policy promoting smaller car use lowers CO_2_ and NO_x_ emissions by decreasing fuel consumption, but this may encourage more driving, or other carbon intensive activities, such as flying [74]. However, this effect should lessen as all sectors of the economy decarbonise, and is difficult to quantify.

This study considered emissions from tailpipes for ICE vehicles and the electrical grid for charging BEVs. It did not consider embedded emissions from the manufacture and supply chains of any vehicle. It is thought that BEVs have greater manufacturing emissions, but lifetime emissions are much lower [51] and will shrink as economies decarbonise. However, for any given fuel type, larger and heavier vehicles are likely to have larger life-cycle emissions than smaller and lighter cars, as fewer materials are used [31].

The public health dangers of SUVs are not just limited to CO_2_ and NO_x_. Particulate matter, noise, and injury risk to other road users all potentially increase with vehicle size and weight [35,36,37]. Particulate matter is excluded in this investigation due to the lack of publicly available data for passenger vehicles registered in the UK. Separating the health effects of NO_x_ and particulate matter is controversial and uncertain, as the pollutants are often highly correlated [7]. This is partly controlled for in this study by the use of unadjusted NO_x_ coefficients, with the assumption that actions that reduce NO_x_ may also reduce particulate matter. There is growing scrutiny on non-exhaust particulate emissions, such as those from brakes, tires, and road wear, especially as the trend towards heavier vehicles, including SUVs and electric vehicles, is likely to result in higher levels of these emissions [75]. However, comprehensive data for these variables are not publicly available for analysis.

Finally, this study assumes a car-centric future to 2050, with continued annual new car sales of 2.5 million per year, in line with industry projections [48]. Although it was not the focus of this work, it is likely that the greatest health benefits could be achieved by shifting to active and public transportation systems [44,76], whilst shared mobility could reduce vehicle life-cycle emissions per traveller [77]. However, the benefits of downsizing SUVs identified in this analysis could occur alongside, and compliment, a structural shift towards active and shared mobility.

## 5. Conclusions

This modelling study indicates that SUVs contribute a large and disproportionate share to CO_2_ and NO_x_ emissions from passenger vehicles in the UK, mainly related to their greater mass. Future policy actions that reduce the mass of SUVs sold in the UK could lead to major CO_2_ and NO_x_ emission reductions. These could be similar in magnitude to but achieved more quickly than the benefits from the electrification of the passenger vehicle fleet. Such changes would be associated with significant public health benefits from improved air quality and mitigation of climate change over the longer-term. To help enable these changes, current manufacturer regulations could be amended so that vehicle CO_2_ limits are decoupled from vehicle mass and instead based on other measures, such as vehicle footprint. Consumer behaviour change could be achieved by directly taxing vehicle mass to disincentivise heavier SUV purchases.

## Figures and Tables

**Figure 1 ijerph-20-06043-f001:**
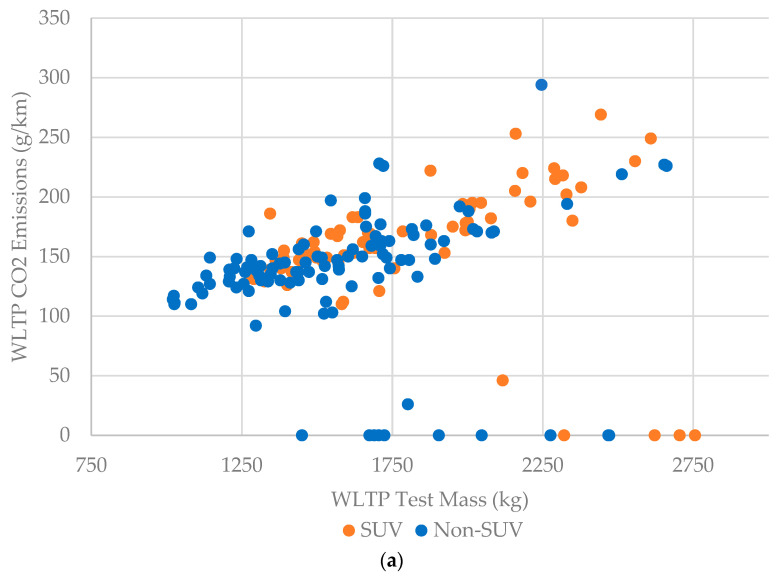
(**a**) Passenger vehicle CO_2_ emissions by mass stratified by SUV status; (**b**) passenger vehicle CO_2_ emissions by mass stratified by fuel type. Note: BEVs have zero tailpipe CO_2_ emissions.

**Figure 2 ijerph-20-06043-f002:**
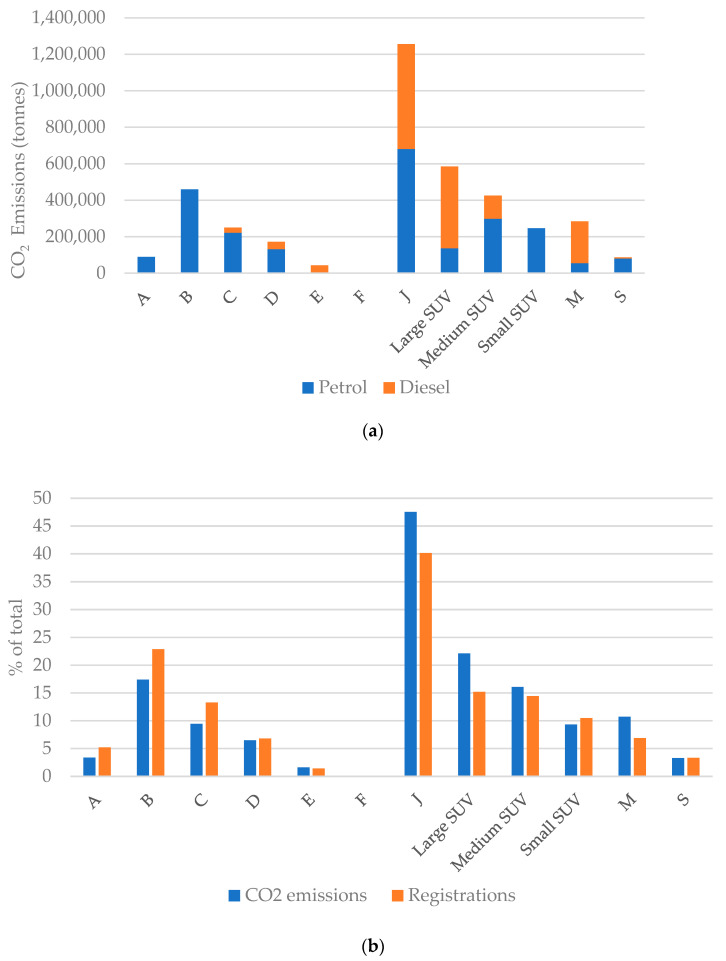
(**a**) Predicted annual CO_2_ emissions for 2020 registrations by fuel type; (**b**) Percentage of annual CO_2_ emissions versus percentage of total registrations in 2020.

**Figure 3 ijerph-20-06043-f003:**
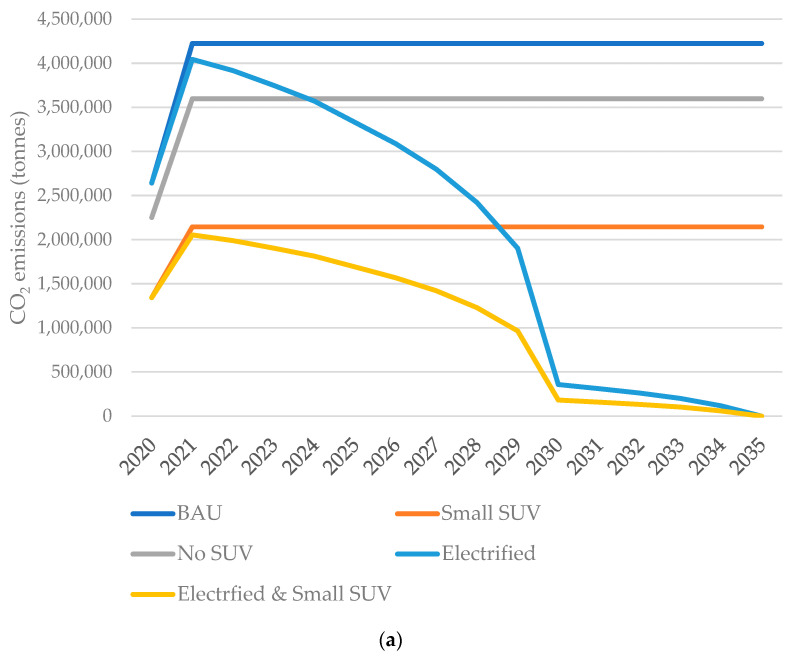
(**a**) Annual CO_2_ emissions from new passenger vehicles. Note the impact of COVID on CO_2_ emissions in 2020 and assumed recovery to pre-pandemic car sales in 2021; (**b**) cumulative CO_2_ emissions of new passenger vehicles from 2020 onwards at 2035 or 2050.

**Figure 4 ijerph-20-06043-f004:**
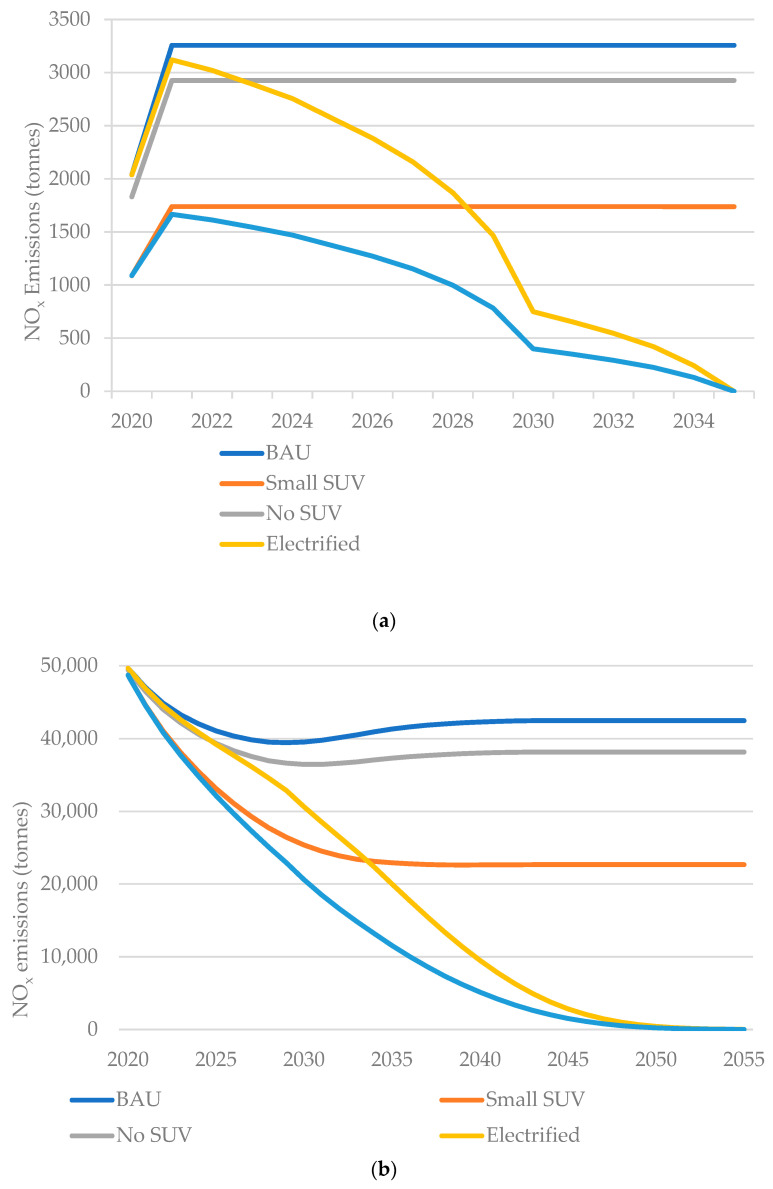
(**a**) NO_x_ emissions in first year of use for new registrations; (**b**) annual NO_x_ emissions from UK passenger vehicle fleet; (**c**) projected NO_2_ decrease relative to the BAU from the UK passenger vehicle fleet.

**Figure 5 ijerph-20-06043-f005:**
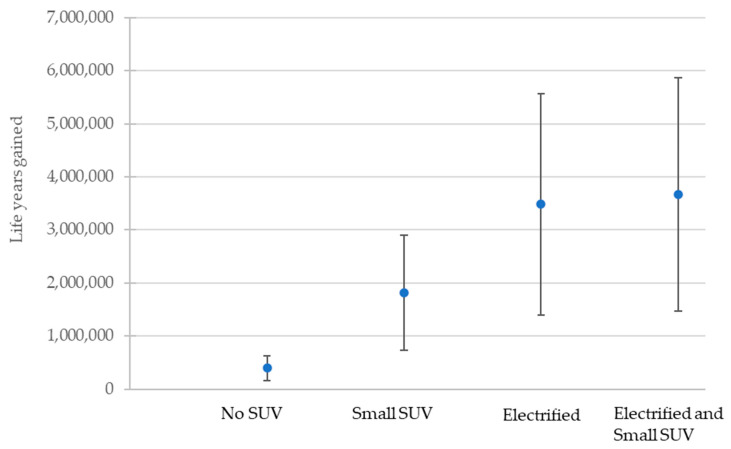
Low, central, and high estimates of life years gained by scenario based on low, central and high NO_2_ coefficients.

**Table 1 ijerph-20-06043-t001:** Overview of EURO segments, other names, and bestselling examples.

EURO Segment	Other Names	Bestselling 2020 UK Generic Model Examples
A	Mini, city car, minicompact	Fiat 500
B	Small, supermini, subcompact	Vauxhall Corsa
C	Medium, small family, compact	Volkswagen Golf
D	Large, large-family,	BMW 3 Series
E	Executive, full-size	BMW 5 Series
F	Luxury, full-size luxury	BMW 7 Series
J	Sports utility vehicles, 4 × 4, crossover	Small SUV: Ford Puma Medium SUV: VW Tiguan Large SUV: Land Rover Defender
M	Multi-purpose	Ford Tourneo
S	Sports	Porsche Taycan

**Table 2 ijerph-20-06043-t002:** Count of vehicle units by EURO segment and fuel type used in CO_2_ model.

EURO Segment	Diesel	Diesel Hybrid	BEV	Petrol	Petrol Hybrid	Petrol PHEV	Row Total	% of Fleet
A			10,351	70,790			81,141	5.2
B			2756	260,112	93,976		356,844	22.9
C	12,917		11,399	116,002	31,164	35,580	207,062	13.3
D	18,035		9081	76,807	2326		106,249	6.8
E	15,691		3130	2948			21,769	1.4
F	1186						1186	0.1
J	194,116	1130	31,146	353,037	42,172	4932	626,533	40.2
Large SUV	141,705		31,146	40,576	18,893	4932	237,252	15.2
Medium SUV	52,411	1130		148,526	23,279		225,346	14.4
Small SUV				163,935			163,935	10.5
M	72,263			35,292			107,555	6.9
S	2285		4836	44,992			52,113	3.3
Column totals	316,493	1130	72,699	959,980	169,638	40,512	1,560,452	
% of fleet	20.3	0.1	4.7	61.5	10.9	2.6		

**Table 3 ijerph-20-06043-t003:** Results of multiple linear regression for WLTP CO_2_ emissions. The overall regression was statistically significant (R^2^ = 0.8857, F = 276.9, *p* < 0.0001).

	Coefficient	95% CI	*p*-Value
WLTP test mass (kg)	0.1028	0.090, 0.116	<0.001
Engine capacity (cm^3^)	0.0198	0.015, 0.025	<0.001
Wheelbase (mm)	−0.0793	−0.10, −0.582	<0.001
Fuel type (1 = Diesel, 0 = Petrol)	−24.366	−29.92, −18.81	<0.001
Constant	181.546	137.7, 225.4	<0.001

**Table 4 ijerph-20-06043-t004:** Regression results for RDE NO_x_ emissions for diesel and petrol vehicles. Diesel: R^2^ = 0.513, F = 12.59, *p* = 0.0003. Petrol: R^2^ = 0.122 F = 4.13, *p* = 0.0228.

	Diesel			Petrol		
	Coefficient	95% CI	*p*-Value	Coefficient	95% CI	*p*-Value
WLTP test mass (kg)	0.0677	0.028, 0.108	0.002	0.0677	0.028, 0.108	0.002
Engine capacity (cm^3^)	0.0299	0.004, 0.056	0.026	0.0299	0.004, 0.056	0.026
Constant	−57.04	−140.54, 26.46	0.17	−57.04	−140.54, 26.46	0.17

**Table 5 ijerph-20-06043-t005:** CO_2_ and NO_x_ emissions ratios (F and E segment omitted as only represented by two or fewer car models).

EURO Segment	Vehicles Registered in 2020 (*n*)	CO_2_/ Registration Ratio	NO_x_/ Registration Ratio	CO_2_ Emissions/Vehicle (Tonnes/Year)	NO_x_ per Vehicle (kg/pa)
A	35,716	0.74	0.599	1.25	0.816
B	220,225	0.77	0.937	1.30	1.276
C	174,629	0.76	0.941	1.28	1.283
D	74,633	1.04	0.734	1.76	1.000
*J*	443,713	1.25	1.123	2.11	1.530
*Large SUV*	114,378	1.68	1.515	2.84	2.064
*Medium SUV*	187,901	1.12	1.048	1.89	1.428
*Small SUV*	141,434	0.89	0.905	1.50	1.232
*M*	19,267	1.56	1.317	2.64	1.794
S	11,187	1.09	0.553	1.85	0.754

**Table 6 ijerph-20-06043-t006:** Cumulative CO_2_ by 2035 or 2050.

Scenario	Cumulative CO_2_ Emitted (Mt)		% Change from BAU	
	2020 to 2035	2020 to 2050	By 2035	By 2050
BAU	542	1,429	0	0
Small SUV	276	727	−49	−49
No SUV	462	1217	−15	−15
Electrified	380	473	−30	−67
Electrified and small SUV	197	248	−63	−83

**Table 7 ijerph-20-06043-t007:** Value of CO_2_ abated by 2035 and 2050 (GBP billions at 2020 prices).

	At 2035			At 2050		
	Low (GBP 151.1/t)	Central (GBP 302.3/t)	High (GBP 453.4/t)	Low (GBP 189.2/t)	Central (GBP 378.3/t)	High (GBP 567.5/t)
Small SUV	40	80	121	133	266	398
No SUV	12	24	36	40	80	120
Electrified	24	49	73	181	362	543
Electrified and small SUV	52	104	156	224	447	671

**Table 8 ijerph-20-06043-t008:** Central estimates of life years and valued gained using HM Treasury’s recommended discount and VOLY.

Scenario	Life Years Gained vs. BAU	% Change from Electrified Scenario	Value (GBP 000,000s)	% Change from Electrified Scenario
No SUV	393,455	−88.7	10,052	−87.9
Small SUV	1,810,500	−50.7	46,255	−48.9
Electrified	3,482,183	0	82,937	0
Electrified and Small SUV	3,669,252	5.4	90,524	9.1

**Table 9 ijerph-20-06043-t009:** Summary of the electrified and small SUV sensitivity analysis (values in GBP 000,000s).

VOLY	Discount Rate	NO_2_ Coefficient (95% CI)		
		Lower Limit (1.008)	Central (1.023)	Upper Limit (1.037)
GBP 20,000	1.5%	10,574	30,175	48,211
	3.50%	3889	11,099	17,732
GBP 60,000	1.5%	31,721	90,524	144,633
	3.50%	11,667	33,296	53,195

## Data Availability

Publicly available datasets were analysed in this study. The datasets used to develop the underlying model can be found here: Vehicle licensing statistics data tables. Available online: https://www.gov.uk/government/statistical-data-sets/vehicle-licensing-statistics-data-tables (accessed on 1 March 2023); for fuel and emissions information: Vehicle Certification Agency. Directgov: https://carfueldata.vehicle-certification-agency.gov.uk/downloads/default.aspx (accessed on 1 March 2023); for vehicle characteristics: https://www.eea.europa.eu/en/datahub (accessed on 1 March 2023).

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
