# Peer review of "Sports Utility Vehicles: A Public Health Model of Their Climate and Air Pollution Impacts in the United Kingdom"

_ijerph, 2023, doi:10.3390/ijerph20116043_

Round 1

Reviewer 1 Report

This paper investigated the correlations between vehicle characteristics and emissions using multiple linear regression. I think this paper is an interesting topic. However, before it can be accepted, it requires major revisions to address the following issues:

1. If the paper does not have a dedicated literature review section and the author does not want to add one, it is recommended to provide a brief literature review in the introduction. This can help identify knowledge gaps, demonstrate the novelty of the research, and justify the problem being addressed, ultimately increasing the paper's academic value.

2. The empirical part is too simple, only 1 year’ data is included in the study design. If possible, it is recommended to test the robustness of the research hypothesis. Relying on only one year of data to predict trends for the next 15 years may be questionable in terms of accuracy. Additionally, the paper overemphasizes data analysis while lacking theoretical analysis to explain the underlying reasons behind the data.

3. The logical connection between paragraphs in the fourth section of the paper is not very strong, and it is suggested that the organization be further refined.

4. The references need to be enriched as much as possible, and repeated citations of the same document should be reduced. The reference format needs to be standardized.

5. Please check the full text and pay attention to some details. For example, there is not “.” at the end of the first paragraph of chapter 1.2.

6. It is recommended to pay attention to the layout of Table 1 of Appendix B, where page numbers are intermingled with table contents.

Author Response

On behalf of the co-authors, I would like to thank you for your thoughtful comments on the manuscript. These have been very helpful in improving the manuscript, especially in respect of the introduction and discussion.

1. If the paper does not have a dedicated literature review section and the author does not want to add one, it is recommended to provide a brief literature review in the introduction. This can help identify knowledge gaps, demonstrate the novelty of the research, and justify the problem being addressed, ultimately increasing the paper's academic value.

Thank you – please find a reworked introduction that includes a more detailed literature review of related work (lines 29-101). However, the impact of SUVs on pollution is understudied, and there is an absence of literature investigating this directly, highlighting the novelty of this work.

2. The empirical part is too simple, only 1 year’ data is included in the study design. If possible, it is recommended to test the robustness of the research hypothesis. Relying on only one year of data to predict trends for the next 15 years may be questionable in terms of accuracy. Additionally, the paper overemphasizes data analysis while lacking theoretical analysis to explain the underlying reasons behind the data.

Thank you – we have included ‘2.1 Conceptual Framework’ to provide a clearer overarching explanation of the NOx and CO2 model (lines 103-131).
The NOx model includes emissions from all vehicles on the road, including those registered in previous years and so relies on many years of data. However, the emissions from vehicles sold in 2020 are assumed to be the same for the business as usual scenario, whose purpose is to serve as a counterfactual to the other policy-relevant scenarios.

It is important that the data for new vehicles only included those sold in 2020 for three reasons:
1. because of the introduction of Euro 6d at the end 2019

2.  conceptually we wanted to use the cleanest vehicles being purchased before the introduction of any policies as the baseline rather than an average over last few years

3. At fleet level vehicle models come and go regularly, preventing averaging across multiple years.

Thank you also for your point on theoretical analysis. We hope adding a new section on Conceptual framework in the methods helps. In addition, we have reorganised and extended the discussion to explore the underlying reasons behind the data in more depth (lines 526-623).

3. The logical connection between paragraphs in the fourth section of the paper is not very strong, and it is suggested that the organization be further refined.

As above, following your comments we have restructured and extended the discussion as you suggest (lines 426-623) Thank you.

4. The references need to be enriched as much as possible, and repeated citations of the same document should be reduced. The reference format needs to be standardized.

A thorough literature review has now enriched the reference list and repeated citation of the same document reduced. Formatting has been updated. Thank you.

5. Please check the full text and pay attention to some details. For example, there is not “.” at the end of the first paragraph of chapter 1.2.

Thank you, we have reviewed the text for similar omissions.

6. It is recommended to pay attention to the layout of Table 1 of Appendix B, where page numbers are intermingled with table contents.

The information in the appendixes have now been brought into the main text following comments from another reviewer.

Reviewer 2 Report

In this manuscript, authors tried to investigate the climate and air pollution impacts of SUVs in the United Kingdom. However, the simulation model used in this research only contains two very simple equations (emission calculation and relative risk calculation). There is not any prediction, optimization, or validation calculations involved. The two simple equations used in this research can only generate very simplified results and the generated results are heavily based on current available data from references. Because there is almost no modeling (due to only using two very simple equations) in this research, the entire manuscript is actually only analyzing current available data from references.

In addition, authors should address the following issues:

1. Authors stated that "This modeling study indicates that SUVs contribute a large and disproportionate share to CO2 and NOx emissions from passenger vehicles in the UK, mainly related to their greater mass. (Line 525-527)" However, this statement is not completely true. SUVs have higher emissions is mainly due to worse aerodynamics, because the relative larger height and width of the vehicle body and wider tires make the drag coefficient high. 

2. There is a typo in Table 1. It should be "BMW 5 Series", not "BMW E Series" as listed in the table.

3. Reference numbering should be added to the tables and information currently in the Appendix. Also, tables and information currently in the Appendix should be moved into main text, because they are not large data set so they can fit into the main text very well.

Author Response

In this manuscript, authors tried to investigate the climate and air pollution impacts of SUVs in the United Kingdom. However, the simulation model used in this research only contains two very simple equations (emission calculation and relative risk calculation). There is not any prediction, optimization, or validation calculations involved. The two simple equations used in this research can only generate very simplified results and the generated results are heavily based on current available data from references. Because there is almost no modeling (due to only using two very simple equations) in this research, the entire manuscript is actually only analyzing current available data from references

Thank you for raising this issue surrounding our novel approach. To assist in its explanation, we have now included a new section ‘2.1 Conceptual framework’ to explain the overarching model (lines 103-131). In this work there are various calculations beyond the two you mention:

1.       Multiple linear regression

2.       Stock turnover model based on historical data

3.       Combining multiple databases to make a bottom-up model based on the emissions of nearly every vehicle on UK roads

4.       BEV emissions calculation incorporating grid carbon intensity

5.       The calculations involved in the IOMLIFET life table analysis (survival probability, cumulative survival and expected life years, VOLYs and discount rates) are described by the IOMLIFET creators (Miller and Hurley, 2006).

The model provides projections under various theoretical scenarios that are designed to mimic diverse potential policies. This is done with the aim of making relative comparisons between policy responses. As a result these are projections, rather than predictions and optimisation and validation equations we consider to be beyond the scope of the study.

1. Authors stated that "This modeling study indicates that SUVs contribute a large and disproportionate share to CO2 and NOx emissions from passenger vehicles in the UK, mainly related to their greater mass. (Line 525-527)" However, this statement is not completely true. SUVs have higher emissions is mainly due to worse aerodynamics, because the relative larger height and width of the vehicle body and wider tires make the drag coefficient high.

Thank you, you raise an interesting point on the relative sources of energy loss for a vehicle. We have reviewed the literature in more depth and have added to the manuscript that SUVs are likely to have reduced fuel efficiency due to poorer aerodynamics.

However, the consensus in the literature suggests that vehicle mass is the dominant explanatory variable for fuel efficiency in cars under typical consumer uses(Burgess 2003, Tolouei 2009). It is especially important in overcoming inertial forces during acceleration and also increasing rolling resistance and therefore is especially dominant in low velocity and start-stop driving. Aerodynamic drag becomes more important at higher speeds such as motorway driving or higher (Howey 2010). We have now explained this in the text.

Interestingly previous studies have found aerodynamic drag is highly correlated with vehicle mass, and added very little to regression models (Tolouei 2009). We were unable to investigate this, as aerodynamic drag data is not available for the UK fleet.

However, the overall thrust of the argument is that SUVs have substantially worse fuel efficiency, even if it is not possible to fully disentangle how much is related to worse aerodynamic drag versus mass.  

2. There is a typo in Table 1. It should be "BMW 5 Series", not "BMW E Series" as listed in the table.

Thank you, this has been amended (line 154) and the manuscript carefully checked for further typos.

3. Reference numbering should be added to the tables and information currently in the Appendix. Also, tables and information currently in the Appendix should be moved into main text, because they are not large data set so they can fit into the main text very well.

Thank you, all material in the appendix has been integrated with the text. Where duplication of material occurred, material has been edited. Please note the data in appendix A (now line 357) has been changed to show unit numbers of vehicles, which is what is of importance to the reader, instead of number of different vehicle models. The underlying model remains the same.

Reviewer 3 Report

In the introduction, I think you should cite other authors who have done work similar to yours, and also other articles that review the effects of these gaseous emissions on human health.

Method. As you mentioned the average NOx emission, I suggest putting the individual fuel consumption of each vehicle (or at least the average)

Results. These graphs in Figures 3 and 4 cannot be straight lines. The correct thing would be to increase emissions over time, since statistical forecasts reveal that there will be a percentage increase in population. So, according to official data, the UK has 519 vehicles per 1000 inhabitants and consequently, emissions will also increase proportionately. That is, the use of electric vehicles will further reduce emissions as the number of years increases. I think it was interesting that you inserted that into the discussion. There are also cost reductions with hospital treatment for these people (and others who would not die).

Author Response

In the introduction, I think you should cite other authors who have done work similar to yours, and also other articles that review the effects of these gaseous emissions on human health.

Thank you – please find a reworked introduction that includes a more detailed literature review of related work (lines 29-101).. However, the impact of SUVs on air pollution is in particular understudied, and there is an absence of literature investigating this directly.

Method. As you mentioned the average NOx emission, I suggest putting the individual fuel consumption of each vehicle (or at least the average

Thank you, unfortunately giving an average fuel consumption of the average vehicle by segment is not possible (as detailed Table 2 line 357). This is because within these segments are a mix of diesel, petrol and electrically powered vehicles, for which fuel efficiency units differ and so cannot be averaged. We don’t believe this omission takes away from the manuscript however and the thrust of the paper is interested in the contribution of different vehicle segments to overall emissions at the UK fleet level (reflecting in part different fuel type mixes, and how these change overtime as well as individual energy efficiency).

Results. These graphs in Figures 3 and 4 cannot be straight lines. The correct thing would be to increase emissions over time, since statistical forecasts reveal that there will be a percentage increase in population. So, according to official data, the UK has 519 vehicles per 1000 inhabitants and consequently, emissions will also increase proportionately. That is, the use of electric vehicles will further reduce emissions as the number of years increases. I think it was interesting that you inserted that into the discussion. There are also cost reductions with hospital treatment for these people (and others who would not die).

Thank you, this is an interesting point. We have elected to take a pragmatic approach for several reasons:

1.       Annual sales of approx. 2.5 million vehicles per year is predicted by the industry (Society of Motoring Manufacturers and Traders, 2021)

2.       Fixing future annual sales acts as a control, so that differences in omissions is down to changes between segments or fuel types as specified in the various scenarios

3.       Conceptually, the business as usual scenario provides a counterfactual that all other policy scenarios can be compared against

4.       We are unable to determine a clear link between population growth and car use that could be incorporated, and therefore elected to use industry projections as per point 1. The UK National Travel Survey shows household car access has been broadly static since the 1970s, and the annual number of car trips and distance travelled per capita has been falling since 2000.

You raise an important additional benefit with regard to potential further cost reductions for reduced hospital treatment is relevant and we have included this as an area for further research. Thank you.

Reviewer 4 Report

Comments for the improvement of study are as follows;

Why should we conduct this study? What is the background of the problem?

Expand introduction section with latest articles.

What is the contribution of the study? Please justify the contribution or applicability of this study

Please write 2-3 lines for literature survey mentioning the findings or limitation of the work

What problem you have tried to solve or contribution of this study for big challenges? Please highlight the novelty of this work and specify the limitation if there are some.

Please expand the methodology section to include the relevant and important details which facilitate the reader to understand how this research can be reproduced.

Provide future recommendations after conclusions.

Update formatting of references.

Author Response

Reviewer 4

Why should we conduct this study? What is the background of the problem?

Thank you, we have extensively reworked the introduction to include more detail on the background, aims and objectives (lines 90-101).

Expand introduction section with latest articles.

Whilst the literature is very limited regarding the public health impact of SUV emissions, we have conducted a deeper literature review as part of the introduction, including relevant articles on electrification, and the effect of vehicle mass which are highly relevant to the UK car market at this time (lines 29-101).

What is the contribution of the study? Please justify the contribution or applicability of this study

As above, this is now explicit in the introduction. The sparse literature regarding SUV emissions and their public health impact serves to highlight the novelty of our approach.

Please write 2-3 lines for literature survey mentioning the findings or limitation of the work

A significantly improved strengths and limitations section of our paper is present in the discussion (line 625-674).

What problem you have tried to solve or contribution of this study for big challenges? Please highlight the novelty of this work and specify the limitation if there are some.

As above, the problem and study aim has been explicitly detailed in the introduction and limitations are explicitly detailed in the discussion.

Please expand the methodology section to include the relevant and important details which facilitate the reader to understand how this research can be reproduced.

Thank you – we have included ‘2.1 Conceptual Framework’ to provide a clearer overarching explanation of the NOx and CO2 model (lines 103-131). We have also included the BEV calculation in the main methods, and expanded our explanation of other aspects of the methods (lines 226-260).  

Provide future recommendations after conclusions.

Thank you.  We have included recommendations for further research and proposed various policy responses according to the IJERPH template. However, co-authors from the UKHSA are Civil Servants to the UK Government and it is beyond our remit to propose specific policy recommendations.

Update formatting of references.

Thank you – we have updated formatting of the references.

Round 2

Reviewer 1 Report

The authors have made a comprehensive revision according to the review comments. I have no problem, I suggest receiving it.

Reviewer 2 Report

Authors have carefully revised the manuscript based on reviewer's review report. The manuscript has been much improved. After carefully reading and checking the manuscript in present form, reviewer recommend this manuscript for publication.